# Biophysical modeling and experimental analysis of the dynamics of *C. elegans* body-wall muscle cells

Xuexing Du[1,2,3], Jennifer Crodelle[4], Victor James Barranca[5], Songting Li[1,2,3]*, Yunzhu Shi[6], Shangbang Gao[6]*, Douglas Zhou[1,2,3,7]*

**1** School of Mathematical Sciences, Shanghai Jiao Tong University, Shanghai, China, **2** Institute of Natural Sciences, Shanghai Jiao Tong University, Shanghai, China, **3** Ministry of Education Key Laboratory of Scientific and Engineering Computing, Shanghai Jiao Tong University, Shanghai, China, **4** Department of Mathematics and Statistics, Middlebury College, Middlebury, Vermont, United States of America, **5** Department of Mathematics and Statistics, Swarthmore College, Swarthmore, Pennsylvania, United States of America, **6** Key Laboratory of Molecular Biophysics of the Ministry of Education, College of Life Science and Technology, Huazhong University of Science and Technology, Wuhan, China, **7** Shanghai Frontier Science Center of Modern Analysis, Shanghai Jiao Tong University, Shanghai, China

* songting@sjtu.edu.cn (SL); sgao@hust.edu.cn (SG); zdz@sjtu.edu.cn (DZ)

**Data availability statement:** All code used for model fitting and plotting is available on a

## Abstract

This study combines experimental techniques and mathematical modeling to investigate the dynamics of C. elegans body-wall muscle cells. Specifically, by conducting voltage clamp and mutant experiments, we identify key ion channels, particularly the L-type voltage-gated calcium channel (EGL-19) and potassium channels (SHK-1, SLO-2), which are crucial for generating action potentials. We develop Hodgkin-Huxley-based models for these channels and integrate them to capture the cells' electrical activity. To ensure the model accurately reflects cellular responses under depolarizing currents, we develop a parallel simulation-based inference method for determining the model's free parameters. This method performs rapid parallel sampling across high-dimensional parameter spaces, fitting the model to the responses of muscle cells to specific stimuli and yielding accurate parameter estimates. We validate our model by comparing its predictions against cellular responses to various current stimuli in experiments and show that our approach effectively determines suitable parameters for accurately modeling the dynamics in mutant cases. Additionally, we discover an optimal response frequency in body-wall muscle cells, which corresponds to a burst firing mode rather than regular firing mode. Our work provides the first experimentally constrained and biophysically detailed muscle cell model of C. elegans, and our analytical framework combined with robust and efficient parametric estimation method can be extended to model construction in other species.

## Author summary

Despite the availability of many biophysical neuron models of *C. elegans*, a biologically detailed model of its muscle cell remains lacking, which hampers an integrated

GitHub repository at https://github.com/XuexingDu/C.elegans-Muscle. All data used in the paper provided in the S1 Data files.

**Funding:** This work was supported by National Key R&D Program of China 2023YFF1204200 (S.L., D.Z.); National Natural Science Foundation of China with Grant No. 12225109, 12071287 (D.Z.); National Natural Science Foundation of China Grant 12271361, 12250710674 (S.L.); Lingang Laboratory Grant No. LG-QS-202202-01, and the Student Innovation Center at Shanghai Jiao Tong University (X.D., S.L., and D.Z.). National Natural Science Foundation of China with Grant No. 32371189 (S.G.), the Major International (Regional) Joint Research Project (32020103007 to S.G.). Science and Technology Commission of Shanghai Municipality under Grant No. 24JS2810400 (X.D., S.L., D.Z.). The funders had no role in study design, data collection and analysis, decision to publish, or preparation of the manuscript.

**Competing interests:** The authors have declared that no competing interests exist.

understanding of the motion control process. We conduct voltage clamp and mutant experiments to identify ion channels that influence the dynamics of body-wall muscle cells. Using these data, we establish Hodgkin-Huxley-based models for these ion channels and integrate them to simulate the electrical activity of the muscle cells. To determine the free parameters of the model, we develop a simulation-based inference method with parallel sampling that aligns the model with the muscle cells' responses to specific stimuli. Our method allows for swift parallel sampling of parameters in high dimensions, facilitating efficient and accurate parameter estimation. To validate the effectiveness of the determined parameters, we verify the cells' responses under different current stimuli in wild type and mutant cases. Furthermore, we investigate the optimal response frequency of body-wall muscle cells and find that it exhibits a frequency consistent with burst firing mode rather than regular firing mode. Our research introduces the first experimentally validated and biophysically detailed model of muscle cells in *C. elegans*. Additionally, our modeling and simulation framework for efficient parametric estimation in high-dimensional dynamical systems can be extended to model constructions in other scenarios.

## Introduction

The nematode *C. elegans* provides a highly accessible model due to its relatively simple neuronal network, consisting of far fewer neurons and synapses compared to more complex organisms [1]. In addition, due to its genetic simplicity and the ease of performing gene manipulations, this species also serves as an ideal platform for studying specific genes or mutations within a simple system, such as its locomotory circuits [2–4]. Furthermore, the connectome of C. elegans has been comprehensively mapped at the cellular level [2,5], providing a valuable and unique resource for both experimental investigations and neuronal network modeling. Despite its simplicity, the *C. elegans* neuronal network possesses remarkable computational efficiency and versatility. This enables the nematode to execute a diverse array of behaviors, including movement, feeding, sleeping, and mating [6–9]. Moreover, this species is capable of adapting its behavior in response to environmental changes and various stimuli, such as hunger, sex, or stress [7,10,11]. These characteristics make *C. elegans* an invaluable model for researching specific neural systems—such as motor, feeding, and olfactory systems—and for understanding the specific cellular elements involved in these functions.

The long-standing perception in neuroscience posits that nematodes are exceptional in their absence of neuronal action potentials [12]. This has been primarily shaped by early electrophysiological studies conducted on the parasitic nematode *Ascaris suum* [13]. These studies demonstrate that the motor neurons of *Ascaris suum* exhibit only graded electrical properties and synaptic transmission, without any evidence of the typical action potentials observed in many other species [12–15]. Additionally, the *C. elegans* genome lacks voltage-gated sodium channel genes [16], a feature not commonly seen in many species, which is thought to correlate with the absence of action potentials in nematode neurons. However, recent discoveries have challenged this notion by revealing the presence of digital signals within *C. elegans*. Surprisingly, neuroscientists have detected calcium-mediated spikes in *C. elegans*, exhibiting features resembling the hallmark characteristics of action potentials seen in other species [12, 15,17–26]. Notably, these have been observed in specific interneurons [24], and various

sensory and motor neurons [12,25,26], with pharyngeal and body-wall muscles also displaying calcium-dependent action potentials at frequencies between 3–10 Hz [17,23,27].

Concurrently, the development of computational models has significantly advanced the understanding of *C. elegans* neuronal physiology. One area of focus is macroscopic network modeling, which has yielded several important insights. For example, a forward locomotion modeling has revealed that AVB premotor interneurons can induce bifurcations in B-type motor neuron dynamics [28]. Furthermore, proprioceptive interactions among neighboring B-type motor neurons synchronize the frequency of body movements [9,28,29]. Meanwhile, detailed electrophysiological single-neuron models are essential for elucidating neuronal transmission and electrical responses at the molecular level. For instance, an olfactory neuron AWA has been shown to encode natural odor stimuli through regenerative all-or-nothing action potentials [12]. These detailed models are also crucial for investigating the roles of specific ion channels in various cells. Notably, T-type calcium currents facilitated by CCA-1 channels play a critical role in eliciting the depolarization of the motor neuron RMD [25]. Similarly, the electrophysiological model of pharyngeal muscle cells has demonstrated that the strong hyperpolarization after each spike is mediated by potassium channels EXP-2 [26,30,31]. Despite the availability of many biophysical neuron models of *C. elegans*, a biologically detailed model of its muscle cell remains lacking, which hampers an integrated understanding of intellectual behaviors of motion control. Therefore, there remains a need for detailed mathematical models that accurately characterize the electrophysiological data of these body-wall muscle cells.

In this study, we perform electrophysiological experiments and develop Hodgkin-Huxley type models to capture the underlying mechanism for action potential generation for body-wall muscle cells [32–35]. Specifically, utilizing voltage clamp techniques, we construct detailed current dynamics for each ion channel based on our experimental data. These ion currents are subsequently used to develop a well-constrained biophysical model of electrical activity in body-wall muscle cells. To determine the free parameters in the model, we develop a parallel simulation-based inference method to fit the responses of body-wall muscle cells under specific current stimuli [36–38]. This method is based on a Bayesian framework and can efficiently explore high-dimensional parameter spaces. It identifies high-probability regions of parameter space that are consistent with experimental data, thereby quantifying parameter uncertainty. We then validate the model's accuracy by comparing its responses to various current stimuli with corresponding experimental data. Additionally, our approach effectively determines suitable parameters for accurately modeling the dynamics in channel mutants and responses in sodium-ion-free solutions. We also explore the optimal response frequency of body-wall muscle cells, finding that it corresponds to a burst firing mode rather than a regular firing mode. Our modeling approach and framework provide detailed current dynamics for each ion channel and facilitate efficient parametric estimation in high-dimensional dynamical systems, which can be extended to model construction of different cell types in other species.

The structure of this paper is organized as follows: Section 2 details the electrophysiology experimental setup and explains the use of voltage clamp experimental data to establish the corresponding ion channel model. It then introduces an algorithm designed to efficiently search high-dimensional parameter spaces to determine the model's optimal parameters. The results obtained are analyzed in Section 3 from both numerical and biological perspectives. Section 4 concludes with a summary and discussion, as well as possible future research directions.

## Materials and methods

### Electrophysiology

Recordings from dissected *C. elegans* body-wall muscles were conducted following established protocols  [23]. Specifically, adult hermaphrodites aged one day were immobilized on slides with adhesive, and the body-wall muscles were exposed through lateral incisions. We then assessed the integrity of the anterior ventral body muscle and the ventral nerve cord using differential interference contrast (DIC) microscopy. Muscle cells were subsequently patched using fire-polished borosilicate pipettes with a resistance of 4-6 MΩ (World Precision Instruments, USA). We recorded membrane currents and potentials in a whole-cell configuration using a Digidata 1440A and a MultiClamp 700A amplifier, coupled with Clampex 10 software for acquisition and Clampfit 10 for data processing (Axon Instruments, Molecular Devices, USA). The data were digitized at a rate of 10-20 kHz and filtered at 2.6 kHz. Using Clampex, we determined cell resistance and capacitance by administering a 10-mV depolarizing pulse from a holding potential of –60mV, enabling the calculation of $Ca^{2+}$ and $K^+$ current densities (pA/pF). Leak currents were not subtracted in these measurements. For recording membrane potentials and $K^+$ currents: The pipette solution contains (in mM): K-gluconate 115; KCl 25; $CaCl_2$ 0.1; $MgCl_2$ 5; BAPTA 1; HEPES 10; $Na_2$ATP 5; $Na_2$GTP 0.5; cAMP 0.5; cGMP 0.5, pH7.2 with KOH, ~320 mOsm. The extracellular solution consists of (in mM): NaCl 150; KCl 5; $CaCl_2$ 5; $MgCl_2$ 1; glucose 10; sucrose 5; HEPES 15, pH7.3 with NaOH, ~330 mOsm. For recording voltage-dependent $Ca^{2+}$ currents: The pipette solution contained (in mM ): CsCl 140; TEA-Cl 10; $MgCl_2$ 5; EGTA 5; HEPES 10, pH7.2 with CsOH, ~320 mOsm. The extracellular solution contained (in mM): TEA-Cl 140; $CaCl_2$ 5; $MgCl_2$ 1; 4-AP 3; glucose 10; sucrose 5; HEPES 15, pH7.4 with CsOH, ~330 mOsm.

While the primary goal of these recordings was to measure membrane currents and potentials under controlled conditions, we observed some minor muscle twitches during the experiments. However, these movements did not significantly impact the data quality. Due to the high electrode seal resistance (above GΩ) between the glass pipette and the cell membrane, there was minimal disruption to the recordings, and no compensation for motion artifacts was necessary. This level of stability ensured that the spontaneous firing we observed reliably reflected the underlying muscle electrical activity.

### Conductance-based model description

In this section, we present the mathematical formulation of our biophysical model. Our model is based on the Hodgkin-Huxley type formulation, which proves to be a powerful computational approach that accurately reproduces the spiking times and membrane voltage waveform of biological neurons in response to current injections [32–35]. The Hodgkin-Huxley type model we construct encompasses a spectrum of ion channels present in *C. elegans*, including the L-type voltage-gated calcium channel EGL-19, voltage-gated potassium channel SHK-1, and $Ca^{2+}$-gated potassium channel SLO-2, along with non-specific passive currents (Leak)  [39,40]. According to the conservation of current, the membrane voltage dynamics can be described by:

$$I_{ext} = C_m \frac{dV}{dt} + I_{total}, \tag{1}$$

where $I_{ext}$ is the external applied current and $I_{total}$ contains all the considered ionic currents

$$I_{total} = I_{EGL-19} + I_{SHK-1} + I_{SLO-2} + I_{Leak}. \tag{2}$$

The ionic currents $I_{\text{EGL-19}}$, $I_{\text{SHK-1}}$, and $I_{Leak}$ are governed by a generalized formulation:

$$I_{\text{ion}} = g_{\text{ion}} m_{\text{ion}}^a h_{\text{ion}}^b \left( V - E_{\text{ion}} \right), \tag{3}$$

where $m_{\text{ion}}$ and $h_{\text{ion}}$ are voltage-dependent activation and inactivation gating variables, respectively. Both gates can be in either an open or closed state. The variables $m_{\text{ion}}$ and $h_{\text{ion}}$ represent the probability of an activation or inactivation gate being in the open state, respectively. For each ion channel, the parameters $a$ and $b$ represent the number of activation and inactivation gates, respectively. The parameter $g_{\text{ion}}$ represents the maximal conductance, while $E_{\text{ion}}$ denotes the reversal potential of the specific ion channel. The gating variables follow the dynamics described by first-order differential equations:

$$\frac{dx}{dt} = \frac{x_{\infty}(V) - x}{\tau_x(V)}, \quad x \in \{m_{\text{ion}}, h_{\text{ion}}\}, \tag{4}$$

where $x_{\infty}(V)$ represents the steady state and $\tau_x(V)$ represents the voltage-dependent time constant.

Meanwhile, there are also ion channels characterized by ligand and voltage regulated currents, e.g., calcium-regulated channel SLO-2. These models are elucidated by specialized models as follows. We modify the model from previous studies [25,41,42] to describe the kinetics of $I_{\text{SLO-2}}$, which requires the binding of two calcium ions to open the channel. The ionic current takes the form of

$$I_{\text{SLO-2}} = g_{\text{SLO-2}} z_{\infty}^3(V) p^2 \left( V - E_K \right)$$

$$\frac{dp}{dt} = \phi \frac{p_{\infty}\left( \left[ \text{Ca}^{2+} \right]_i \right) - p}{\tau_p\left( \left[ \text{Ca}^{2+} \right]_i \right)}, \tag{5}$$

where $\left[ \text{Ca}^{2+} \right]_i$ is the intracellular calcium concentration, and

$$p_{\infty} = \alpha \left[ \text{Ca}^{2+} \right]_i^2 \Big/ \left( \alpha \left[ \text{Ca}^{2+} \right]_i^2 + \beta \right), \quad \tau_p = 1 \Big/ \left( \alpha \left[ \text{Ca}^{2+} \right]_i^2 + \beta \right). \tag{6}$$

Here, $\alpha = 58 \text{ ms}^{-1}\text{mM}^{-2}$ and $\beta = 0.09 \text{ ms}^{-1}$ are rate constants [25,43,44]. Additionally, the parameter $z_{\infty}$ in Eq. 5 is the voltage-dependent equilibrium value.

Due to the dependence of the gating variable $p$ on $\left[ \text{Ca}^{2+} \right]_i$, it is imperative to undertake an estimation of $\left[ \text{Ca}^{2+} \right]_i$. The calculation is performed using the following differential equation:

$$\frac{d\left[ \text{Ca}^{2+} \right]_i}{dt} = -\frac{I_{Ca}}{2FAd} - \gamma \cdot \left( \left[ \text{Ca}^{2+} \right]_i - \left[ \text{Ca}^{2+} \right]_r \right), \tag{7}$$

where $\left[ \text{Ca}^{2+} \right]_r$ is the resting intracellular calcium concentration. The parameter $d$ denotes the depth of a proximal shell adjacent to the cell's surface, which encompasses an area $A$. The Faraday constant $F$ indicates the charge per mole and the parameter $\gamma$ represents the recovery rate for calcium ions [45,46].

We initially establish an estimated range for the model parameters. There are two methods to estimate these parameters. The first method involves fitting experimental voltage clamp data from various ion channels, which helps determine a subset of the model's parameters. Due to individual cell variability, the model parameters for different cells exhibit a range of values. Since voltage clamp experiments for different ion channels are conducted on different

cells, solely fitting the voltage clamp data to determine all parameters does not accurately capture the cells' voltage responses to current injection, which is the primary focus of our model. Consequently, we consider the maximum conductance of each ion channel and the cell membrane capacitance as free parameters. The initially estimated range of these parameters is then incorporated into a parallel simulation-based inference technique to estimate the probability distributions of parameters, fitting the response of body-wall muscle cells to specific external current stimuli.

## Parallel simulation-based inference

Simulation-based inference (SBI) is a powerful statistical inference approach aimed at estimating parameters of a simulation model based on observed data in experiment [38,47,48]. This method has demonstrated considerable efficacy in numerous real-world applications, spanning a diverse array of scientific domains, including population genetics, neuroscience, epidemiology, climate science, astrophysics, and cosmology [38,47,49–51]. However, the challenge in our study lies in navigating a high-dimensional parameter space, which renders existing algorithms computationally intensive and time-consuming. To address this, we devise a parallelized version of SBI capable of efficiently exploring high-dimensional parameter spaces through GPU acceleration.

Before introducing the algorithm, we first present the experimental dataset used in our study. We generate four spike trains using four constant current stimuli of 15 pA, 20 pA, 25 pA, and 30 pA. One of them is selected as the training dataset, with the remaining three spike trains serving as the test datasets.

**Algorithm 1** Parallel Simulation-Based Inference

1: **Input:**
2:    Observed data: $\mathbf{x}_o$; Prior distribution: $p(\theta)$
3:    Neural network posterior estimator: $q_\psi(\theta|\mathbf{x})$
4:    Number of maximum rounds: $R$
5:    Sample number for each round: $N_r$
6:    Tolerance for convergence: $\epsilon$
7: **Output:** Posterior distribution: $p(\theta|\mathbf{x}_o)$
8: Randomly initialize neural network parameters $\psi$
9: $\tilde{p}_1(\theta) := p(\theta)$
10: $N := 0$
11: **for** $r = 1$ to $R$ **do**
12:    **for** $i = 1$ to $N_r$ **do**
13:       Sample $\theta_{N+i} \sim \tilde{p}_r(\theta)$
14:       Simulate $\mathbf{x}_{N+i} \sim p(\mathbf{x}|\theta_{N+i})$
15:    **end for**
16:    $N \leftarrow N + N_r$
17:    Train neural network $q_\psi(\theta|\mathbf{x}) \leftarrow \arg\min_\psi \sum_{j=1}^{N} \mathcal{L}(\theta_j, \mathbf{x}_j)$
18:    $\tilde{p}_{r+1}(\theta) := q_\psi(\theta|\mathbf{x}_o)$
19:    **if** $D_{KL}(\tilde{p}_{r+1}(\theta) \| \tilde{p}_r(\theta)) < \epsilon$ **then break**
20: **end for**
21: **return** $p(\theta|\mathbf{x}_o) \leftarrow q_\psi(\theta|\mathbf{x}_o)$

Our algorithm is outlined in Algorithm 1. Based on the experimental data, we provide approximate intervals for the model's parameter values, choosing a uniform prior distribution $p(\theta)$ within these intervals. We then extract the essential features from the experimental data, as shown in Table 1, which we refer to as the observed data $\mathbf{x}_o$. Our objective is to

Table 1. **Brief explanation of model summary statistics.** AP = **action potential;** $V$ = **membrane potential.** On the left, we detail the statistics utilized for training all model posteriors. On the right, we describe an additional set of statistics exclusive to the baseline posterior estimate. This baseline is employed for method comparison in Table 2.

| Statistics set | | Additional set | |
|---|---|---|---|
| Feature | Explanation | Feature | Explanation |
| APC | AP count | $APC\,(T_{1/4})$ | AP count of first 1/4 |
| latency | Latency of $1^{\text{st}}$ AP | APA | Amplitude of $1^{st}$ AP |
| $\mu\,(V)$ | Mean of $V$ | ISI | Inter-spike-interval |
| $\text{Var}\,(V)$ | Variance of $V$ | $CV_{ISI}$ | CV of ISI |
| $\mu\,(V_{\text{rest}})$ | Mean resting $V$ | $\sigma\,(V_{\text{rest}})$ | Standard deviation of resting $V_m$ |

determine a posterior distribution of the parameters $\theta$. This is achieved through the Maximum a Posteriori (MAP) method, which yields the distribution $p(\theta\,|\,\mathbf{x})$. By conditioning on the observed data $\mathbf{x} = \mathbf{x}_o$, we ultimately obtain $p(\theta\,|\,\mathbf{x}_o)$. The algorithm operates over multiple rounds, beginning with the initial round. In this round, with the simulation number $N = N_1$, we sample $N$ parameter values from the prior distribution $p(\theta)$, denoted as $\theta_i \sim p(\theta)$ for $i = 1, 2, \cdots, N$. These sampled values are then used to run simulations, a process known to be time-consuming, especially for large $N$. To mitigate this, we leverage GPU parallel computation to vectorize, parallelize, and utilize just-in-time compilation for the entire simulation process, using the BrainPy software [52,53]. The outputs of the model simulations, specifically the generated voltage curves, are summarized by key features, denoted as $\mathbf{x}_i$ and detailed in Table 1. This summarization step is also vectorized and processed on the GPU. The resulting $N$ parameter-data pairs, represented as $(\theta_i, \mathbf{x}_i)$ for $i = 1, 2, \cdots, N$, are used to train a neural network posterior estimator $q_\psi(\theta\,|\,\mathbf{x})$, where $\psi$ denotes the neural network parameters. The neural network learns the posterior probability based on a masked autoregressive flow (MAF) method [54]. MAF transforms a simple base distribution, typically a Gaussian, into a complex target distribution through a series of autoregressive and invertible transformations. The network parameters $\psi$ are optimized by minimizing the objective function $\sum_{j=1}^{N} \mathcal{L}\left(\theta_j, \mathbf{x}_j\right)$, where

$$\mathcal{L}\left(\theta_j, \mathbf{x}_j\right) = -\log q_\psi\left(\theta_j\,|\,\mathbf{x}_j\right). \tag{8}$$

Finally, the trained neural network posterior estimator: $q_\psi(\theta\,|\,\mathbf{x})$ is applied to the observation data $\mathbf{x}_o$, yielding the posterior distribution $q_\psi(\theta\,|\,\mathbf{x}_o)$. This constitutes the initial round of inference. In subsequent rounds, samples from the obtained posterior distribution conditioned on the observed data, $\tilde{p}_{r+1}(\theta) = q_\psi(\theta\,|\,\mathbf{x}_o)$, are used to simulate a new training set. This new training set is then combined with the previous dataset to retrain the network. This process repeats for a specified number of rounds or until the Kullback-Leibler (KL) divergence [55] between successive posterior distributions falls below a predefined threshold $\epsilon$, indicating convergence. The KL divergence measures how one probability distribution diverges from a second, expected probability distribution, thus a lower value suggests the distributions are more similar. As the key part of our algorithm includes the Expectation-Maximization (EM) algorithm, its convergence can be theoretically guaranteed based on the EM algorithm's convergence, given that the number of simulation trials is sufficiently large. Specifically, as the total number of simulations $N$ increases, the estimated posterior $q_\psi(\theta\,|\,\mathbf{x}_o)$ will eventually converge to the posterior distribution conditioned on the observed data $p(\theta\,|\,\mathbf{x}_o)$ [49,56].

Given the high dimensionality of the parameter space in our study, the algorithm requires a substantial number of model simulations to yield satisfactory results. The original SBI

algorithm, due to computational time and hardware limitations, could only perform a limited number of model simulations in each round. This necessitated multiple rounds for the algorithm to converge. Our parallel SBI algorithm improves upon previous methodologies by utilizing GPU-based vectorization and parallelization, enabling a significant number of model simulations to be performed concurrently in each round within a brief timeframe. This enhancement accelerates the algorithm's convergence, thereby reducing the number of rounds required. Consequently, this reduction in rounds decreases the number of neural network training sessions, thereby lowering computational costs and reducing execution time.

## Results

Our experimental observations indicate that body-wall muscle cell spikes exhibit a stereotypical shape characterized by a fast upstroke, followed by a rapid downstroke and afterhyperpolarization, as shown in Fig 1A. These spikes can display "burst" and "regular" firing modes. In this section, we develop a biologically detailed model of the body-wall muscle cells and investigate the physiological mechanisms underlying our experimental observations. In particular, we analyze the roles of individual ionic currents within the overall cell dynamics.

### Body-wall muscle cells fire all-or-nothing action potentials

To explore the biophysical underlying mechanisms of the *C. elegans* motor circuits, we have conducted an electrophysiological survey of body-wall-muscle cells in *C. elegans*. Classic whole-cell configuration, by using a Digidata 1440A and a MultiClamp 700A amplifier, was made to record the isolated voltage activated $K^+$ currents and voltage-gated $Ca^{2+}$ currents from the muscle cells as shown in Fig 1B.

It is well-established that voltage-dependent potassium channels, triggered by depolarization, play a crucial role in terminating action potentials. Therefore, it is imperative to investigate all potassium channels involved in regulating the electrical activity of body-wall muscle cells. Fig 1C provides a comprehensive overview of the currents associated with voltage-gated potassium channels expressed in *C. elegans*. Mutant voltage clamp currents exhibit significant findings: a marked decrease in current response in *shk-1(lf)* mutants and a moderate reduction in *slo-2(lf)* mutants, and only nominal alterations in other mutant varieties (Figs 1C, 1D and S1 Fig). These observations suggest that the SHK-1 channel plays a central role as the primary voltage-gated potassium channel responsible for repolarizing action potentials, while the SLO-2 channel contributes minimally to this repolarization process.

Our previous work [23] established that the action potentials of *C. elegans* body-wall muscle cells are calcium-dependent. To further explore the primary channels influencing the electrophysiological activity of these muscle cells, we conduct voltage clamp experiments. Because *egl-19* null animals are embryonically lethal, two viable, recessive alleles with partial loss-of-function, *n582* and *ad1006*, are examined. Fig 1E illustrates altered kinetics in *egl-19* mutants, indicating that EGL-19 is responsible for eliciting muscle cell action potentials.

These findings highlight the voltage-dependent $Ca^{2+}$ channel EGL-19, in conjunction with $K^+$ channels SHK-1 and SLO-2, collectively contribute to the generation of action potentials in *C. elegans* body-wall muscle cells.

### Modeling channel dynamics based on experimental data

Based on the previous discussion in Sec. Materials and methods, the parameters for the SHK-1 channel model are determined using voltage clamp experimental data. As shown in Fig 2A, we perform numerical curve fitting for each voltage clamp protocol, where the voltage is held

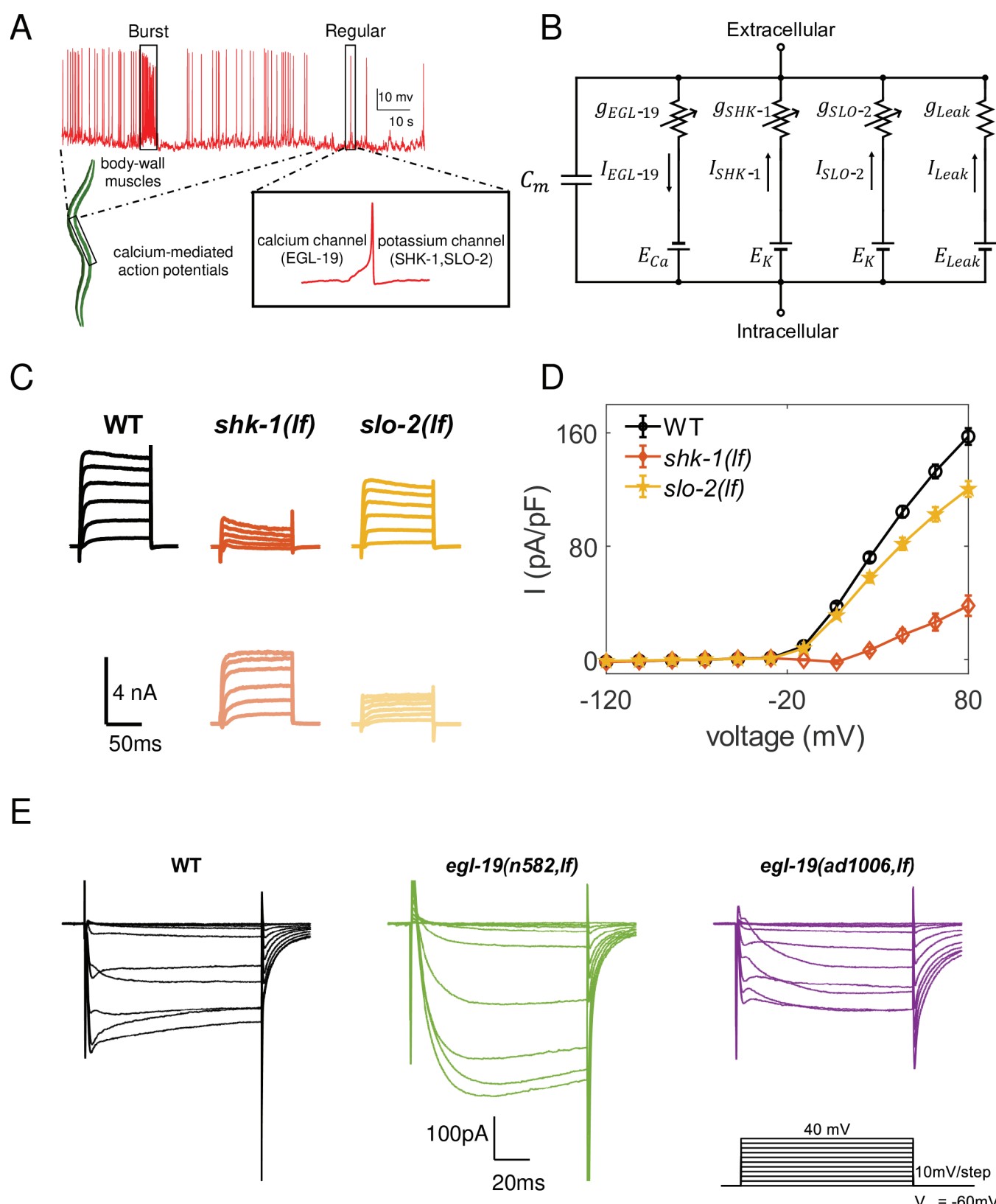

**Fig 1. Electrophysiological characterization of *C. elegans* body-wall muscles.** (A) The graphical representation of *C. elegans* body-wall muscle cells, with their electrical activities recorded from experiments in a wild-type animal. The green schematic is an illustration of the *C. elegans*. The red traces represent an in vivo recording of spontaneous action potentials under baseline conditions, with no external current stimulation (0 pA during recording), showcasing both

"burst" and "regular" firing patterns. A zoomed-in view of a single spike is shown in the inset, highlighting the distinct contributions of these ion channels. (B) An equivalent circuit diagram of the muscle model, illustrating the key electrical components and parameters of the body-wall muscle model. (C) Voltage clamp currents of the potassium channels. The top section of each column illustrates the currents in different mutants, while the bottom section depicts the ionic currents obtained by subtracting the corresponding mutant currents from the total wild type (WT) currents, highlighting a significant decrease in *shk-1(lf)* and a slight reduction in *slo-2(lf)* mutants. (D) Steady-state current density variations in different potassium channel mutants as a function of cell voltage. (E) Voltage clamp calcium currents of wild-type(black) and mutants, specifically *egl-19(n582,lf)* (green) and *egl-19(ad1006,lf)* (purple). The protocol involving voltage steps from -60 mV to +40 mV in 10 mV increments, a holding potential of $V_h = -60$ mV. Details on the mutants are provided in the main text.

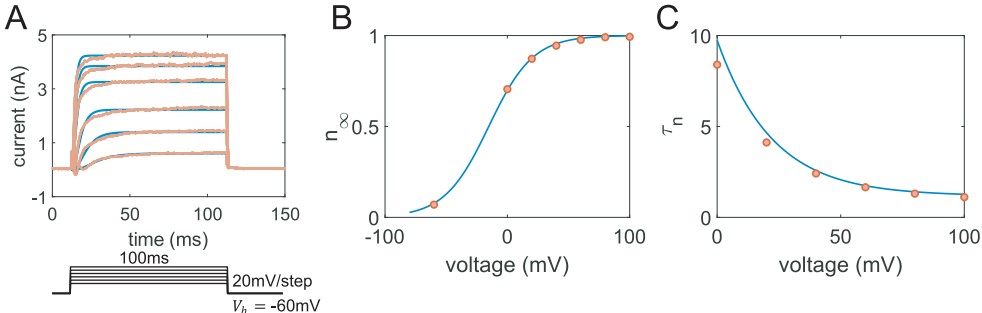

**Fig 2**. **Results of SHK-1 potassium channels model.** (A) Upper Panel: The curve fitting for the SHK-1 channel. Experimental data (wild-type minus SHK-1 mutants) displayed in light red, sourced from *shk-1(lf)* mutants in Fig 1C, with blue lines representing the fitting results. Lower Panel: The protocol involving voltage steps from 0 mV to +100 mV in 20 mV increments, a holding potential of $V_h = -60$ mV, and step duration of 100 ms. (B) The steady-state activation curve alongside the experimental data (illustrated with blue line and red dots) extracted from panel A. (C) The activation time constant function.

constant. Through this process, we obtain appropriate values for $\tau_n$ and $n_\infty$ at specific voltages, with the results presented in Fig 2A. Subsequently, these values are further analyzed to establish the voltage-dependent functions $\tau_n(V)$ and $n_\infty(V)$, as demonstrated in Fig 2B and 2C.

Next, we focus on modeling the calcium channel EGL-19. To address the variability in experimental data for calcium channels across different individuals, we analyze the current-voltage (I-V) relationship of calcium currents from multiple individual cells. We then calculate the mean and standard error, as illustrated in Fig 3B. We determine model parameters based on both voltage clamp experimental data in Fig 1E and the I-V relationship in Fig 3B, with the final results presented in Fig 3A and 3B. Subsequently, we obtain the parameters $\tau_m$, $\tau_h$, $m_\infty$, and $n_\infty$ at specific voltages. Based on these values, we establish the functional forms for the gating variables $m$ and $h$, as illustrated in Fig 3C and 3D.

The calcium-regulated potassium channels SLO-2 are characterized by their intricate dynamics, influenced by both calcium ion concentration levels and voltage amplitude, as modeled previously. Determining numerous parameters solely from voltage clamp data is difficult. Therefore, our methodology predominantly relies on using the parallel SBI method to fit the model to action potential traces observed in body-wall muscle cells subjected to specific electrical stimuli, ensuring precise parameter estimation.

The model of leak current for the *C. elegans* body-wall muscle cells is

$$I_{\text{Leak}} = g_{\text{Leak}} \left( V - E_{\text{Leak}} \right), \tag{9}$$

where $E_{\text{Leak}}$ corresponds to the reversal potential of the channel and $g_{\text{Leak}}$ is the leak conductance that can be estimated in experiment by assuming the cell is a linear integrator [57].

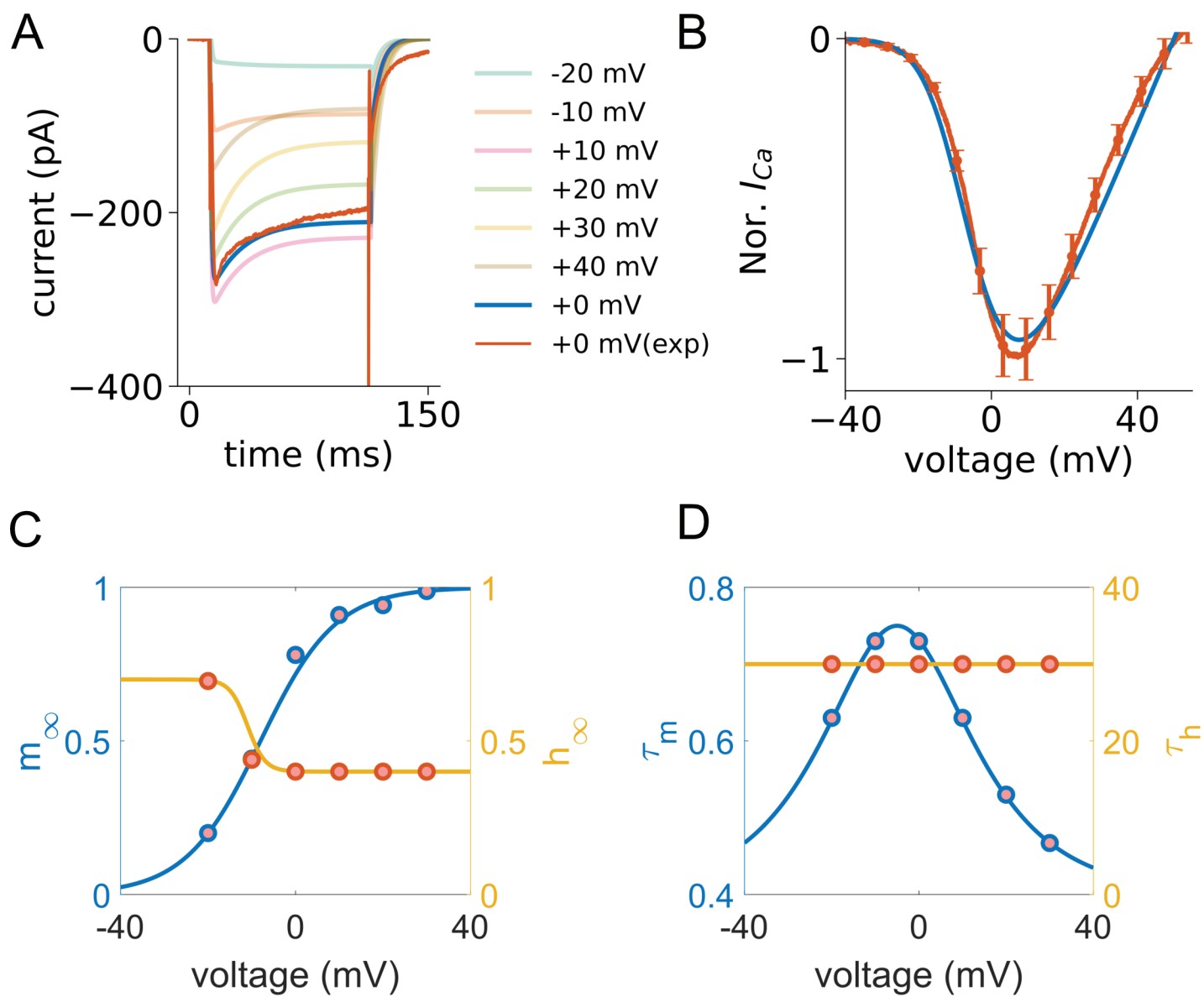

**Fig 3. Results of EGL-19 calcium channel model.** (A) Estimation of calcium currents across voltage steps is represented using distinct colors. The blue line, serving as a representative trace, illustrates the response at a 0 $mV$ voltage clamp, whereas the red line depicts the corresponding experimental trace under the same clamp conditions. (B) The normalized current-voltage relationship for EGL-19, depicting both experimental (red) and simulation (blue) results (Error bar: Standard Error of the Mean). (C) Fitting results for the time constant functions of gating variables. Red dots represent experimental data, while the blue and orange lines represent the gating variables $m$ and $h$, respectively. (D) Steady-state functions of gating variables.

By combining experimental measurements of the four aforementioned ion channels and considering individual cell variability, we can initially estimate the confidence intervals for the corresponding parameters of these ion channels. However, in the case of *C. elegans*, the generation of action potentials may involve additional channels. In the following discussion, we will integrate ion channel candidates mentioned in previous studies on motor neurons and muscle cells, alongside numerical modeling methods, to identify these additional

ion channels. Furthermore, we will use our designed parallel algorithm of simulation-based inference method to accurately determine the parameter ranges for all these channels.

**The resulting 7-dimensional HH type model.** We note that when the model contains only the four previously mentioned ion channels, the action potentials exhibit premature repolarization compared to experimental data, as shown in S2 Fig. Additionally, the resting potential deviates from that observed in the experimental data. To address these issues, we introduce a potassium ionic current, denoted as $K_r$, which serves as an early-phase inhibitory current, and the NCA sodium leak channel, which has a conserved role in determining the neuronal resting membrane potential in our model [58].

The final 7-dimensional HH type model for body-wall muscle cells is given by the following:

$$\begin{cases} I_{ext} = C_m \dfrac{dV}{dt} + I_{\text{SHK-1}} + I_{\text{EGL-19}} + I_{\text{SLO-2}} + I_{Kr} + I_{Na} + I_{\text{Leak}} \\[2mm] \dfrac{dx}{dt} = \dfrac{x_\infty - x}{\tau_x}, \quad x \in \{m, h, n, p, q\} \\[2mm] \dfrac{d\left[\text{Ca}^{2+}\right]_i}{dt} = -\dfrac{I_{\text{Ca}}}{2FAd} - \gamma \cdot \left(\left[\text{Ca}^{2+}\right]_i - \left[\text{Ca}^{2+}\right]_r\right). \end{cases} \quad (10)$$

The six ionic currents in our model are shown in the following:

$$\begin{aligned} I_{\text{SHK-1}} &= g_{\text{SHK-1}} \cdot n^4 \cdot (V - E_K) \\ I_{\text{EGL-19}} &= g_{\text{EGL-19}} \cdot m^2 \cdot h \cdot (V - E_{Ca}) \\ I_{\text{SLO-2}} &= g_{\text{SLO-2}} z_\infty^3(V) p^2 (V - E_K) \\ I_{\text{Kr}} &= g_{Kr}(1-q) q_\infty(V)(V - E_K) \\ I_{Na} &= g_{Na} \cdot (V - E_{Na}) \\ I_{\text{Leak}} &= g_{\text{Leak}} \cdot (V - E_{\text{Leak}}), \end{aligned} \quad (11)$$

where $g_x(x = \text{SHK-1, EGL-19, SLO-2, Kr, Na, Leak})$ and $E_x(x = \text{K, Ca, Na, Leak})$ denote the maximal ionic conductance and the reversal potential for each respective current. Additionally, $n$, $p$ and $q$ correspond to the activation gating variables for $I_{\text{SHK-1}}$, $I_{\text{SLO-2}}$ and $I_{Kr}$, respectively. The $m$ and $h$ represent the activation and inactivation gating variables for $I_{\text{EGL-19}}$, respectively.

As detailed in Sec. Materials and methods, we determine the free model parameters as illustrated in Fig 4E. These parameters include maximal conductance values ($g_{\text{EGL-19}}$, $g_{\text{SHK-1}}$, $g_{\text{SLO-2}}$, and $g_{\text{Leak}}$), membrane capacitance ($C_m$), and voltage shift value ($V_{th}$). For more details on the voltage shift, please refer to S1 Appendix. To measure the differences between the model-generated spike train and the experimental data, we calculate five features of the voltage trace in Fig 4A (red curve), as listed in Table 1. These features provide a comprehensive representation of neuronal activity. Specifically, the action potential count and latency of the first spike indicate neuronal excitability and response speed, respectively, while the mean and variance of the voltage reflect overall activity. The resting potential serves as a baseline indicator of the neuron's physiological state. Furthermore, these features are also suitable for GPU parallelization and vectorized computation, thereby enhancing computational efficiency. Additionally, we utilize our newly-developed SBI method to efficiently explore the high-dimensional parameter space, as described in the Sec. Materials and methods. By leveraging the computational power of A100 GPUs and parallel computing, this approach significantly enhances the speed of parameter sampling in high-dimensional spaces. This method outperforms the original SBI method by two orders of magnitude in terms of runtime and

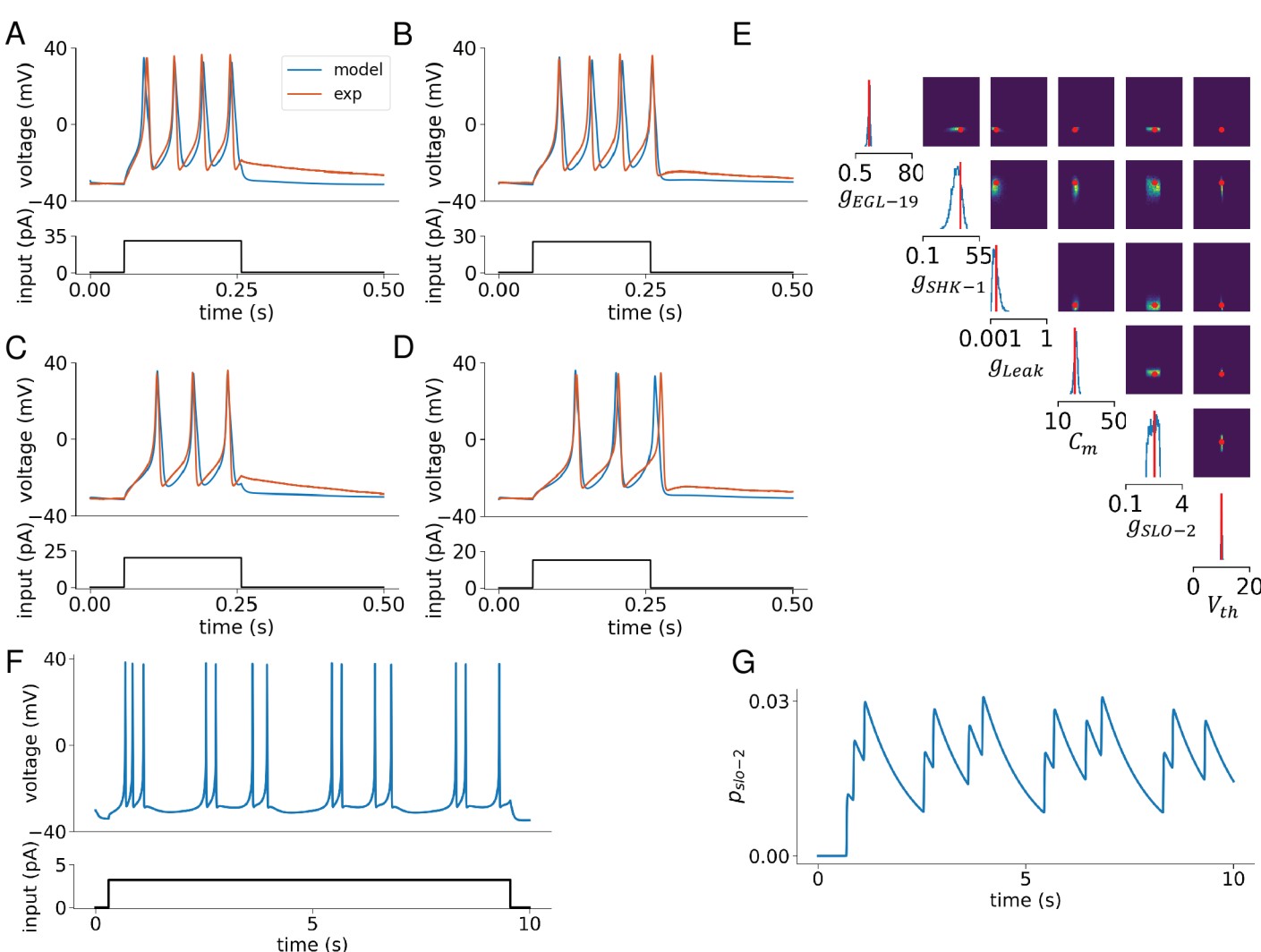

**Fig 4. Detailed results of model fitting in *C. elegans* body-wall muscle cell.** (A–D) Presentation of elicited spike trains across varying stimulation currents from 15 pA to 30 pA, increasing in 5 pA steps. The simulation results are depicted with blue curves, juxtaposed against red curves representing actual experimental data. (E) Cornerplot showing the marginal and pairwise marginal distributions of the 6-dimensional posterior based on five voltage features, including spike count, mean resting potential, time to initial spike, etc. (Table 1). The posterior distribution effectively includes the true parameters within a region of high probability, with the red lines indicating our chosen final values. (F) Spike trains induced by a steady 3.2 pA current, revealing two distinct firing modes in body-wall muscles: "burst" and "regular." (G) The $p_{slo-2}$ gating variable's behavior, which is modulated by both calcium ion concentration and voltage.

also maintains an accuracy comparable to benchmarks, as demonstrated in Table 2. The final results are depicted in S1 Table. The simulated curves now demonstrate consistency in action potential frequency, amplitude, and resting potential when compared to the experimental curves, as shown in Figs 4A–4D.

As illustrated in Fig 4F and 4G, the potassium channel SLO-2 plays a crucial role in modulating two distinct firing modes in body-wall muscle cells. With an increase in calcium ion concentration, SLO-2 inhibits action potentials. Due to its relatively slow activation time, the channel allows the cell to continue firing, resulting in two distinct firing modes under constant current input: the "burst" and "regular" firing modes. These findings are consistent with experimental observations, as shown in Fig 1A.

**Table 2. Performance of Simulation-Based Inference (SBI) and its GPU-parallelized version (mean ± standard deviation across various simulation setups).** Training times are reported in minutes and simulation times, defined as the time required to establish $N$ pairs $(\theta_n, \mathbf{x}_n)$ through model simulation, are reported in seconds. The last column (KL) measures the Kullback-Leibler (KL) divergence to estimate the accuracy of posterior approximations. As no ground truth data is available for our model, the GPU-parallelized version of SBI is employed to estimate the posterior. We utilize a substantially larger set of summary statistics as detailed in Table 1, and the number of samples drawn from the prior distribution has been increased to $2 \times 10^5$. The differences in performance metrics highlight the efficiency gains with GPU parallelization, particularly in reduced simulation times across all simulation counts and dimensions. The mean and standard deviation are obtained across all 10 runs.

| Method | Training Dataset Size (million) | Dimensions | Dataset Preparation (s) | Training Time (min) | KL |
|---|---|---|---|---|---|
| SBI | 0.02 | 6 | 236 ± 30.74 | 3.81 ± 0.18 | 0.32 ± 0.11 |
| | 0.05 | 7 | 1432 ± 132.34 | 22.32 ± 1.12 | 1.21 ± 2.04 |
| | 0.1 | 8 | 5589 ± 765.84 | 49.42 ± 2.64 | 4.62 ± 0.85 |
| SBI (GPU parallel) | 0.02 | 6 | 10.15 ± 2.83 | 2.51 ± 0.14 | 0.31 ± 0.14 |
| | 0.05 | 7 | 12.42 ± 3.01 | 15.08 ± 0.64 | 1.19 ± 0.31 |
| | 0.1 | 8 | 14.98 ± 3.74 | 36.15 ± 1.84 | 4.53 ± 0.62 |

## Prediction of dynamics in mutants and different extracellular solutions

To delve deeper into the dynamical properties of body-wall muscle cells, we investigate how mutants and alterations in extracellular solutions significantly impact the dynamics of our model.

To study the role of *egl-19* in action potential generation, we use two viable, recessive, partial loss-of-function (lf) alleles, *n582* and *ad1006*, as *egl-19* null mutants are embryonically lethal. Both *egl-19(n582)* and *egl-19(ad1006)* are mutations in the same gene, leading to dysfunction of the *egl-19* channel. Despite some differences in their effects on calcium currents, these alleles exhibit similar phenotypes in terms of muscle function, including flaccid paralysis, slow movement, feeble pharyngeal pumping, and defective egg-laying. These shared phenotypic traits have been previously described in the literature [42]. We then use our model to explore the impact of these mutations on action potentials in body-wall muscle cells.

In studying the *egl-19(ad1006,lf)* mutants, a substantial decrease in calcium current amplitudes is observed, as illustrated in Fig 1C. To replicate these experimental findings, our simulation incorporates a reduction in the maximum conductance for the calcium ion channel EGL-19. This modification results in a notable reduction in action potential amplitudes, a finding that our experiments have confirmed. Specifically, under a 30 pA current injection, the average action potential amplitude in *egl-19(ad1006,lf)* mutants is approximately half that of wild-type cells, while the inter-spike interval (ISI) is significantly prolonged, as illustrated in the bar graph in Fig 5B and 5C. Notably, the mutant model agrees with the shape of action potentials under constant current injections, as shown in Fig 5A–5C.

While the *egl-19(ad1006,lf)* mutants exhibit reduced current amplitudes, another aspect of calcium channel dynamics is revealed in the study of *egl-19(n582,lf)* mutants. Namely, experimental data indicate an increase in the activation time constants of the EGL-19 channel, denoted as $\tau_m$ in these mutant models, as shown in Fig 1C. Our simulations, depicted in Fig 5D and 5E, replicate this observation by increasing the $\tau_m$ values several-fold. As shown in Fig 5F, with each incremental increase in $\tau_m$, it approaches $\tau_n$. When $\tau_m$ is increased 20-fold, it surpasses $\tau_n$ throughout most of the firing voltage range. This adjustment significantly reduces the firing rate of the cells compared to wild-type, even leading to a marked inability to generate action potentials, as illustrated in Fig 5D and 5E. Similarly, under the same current

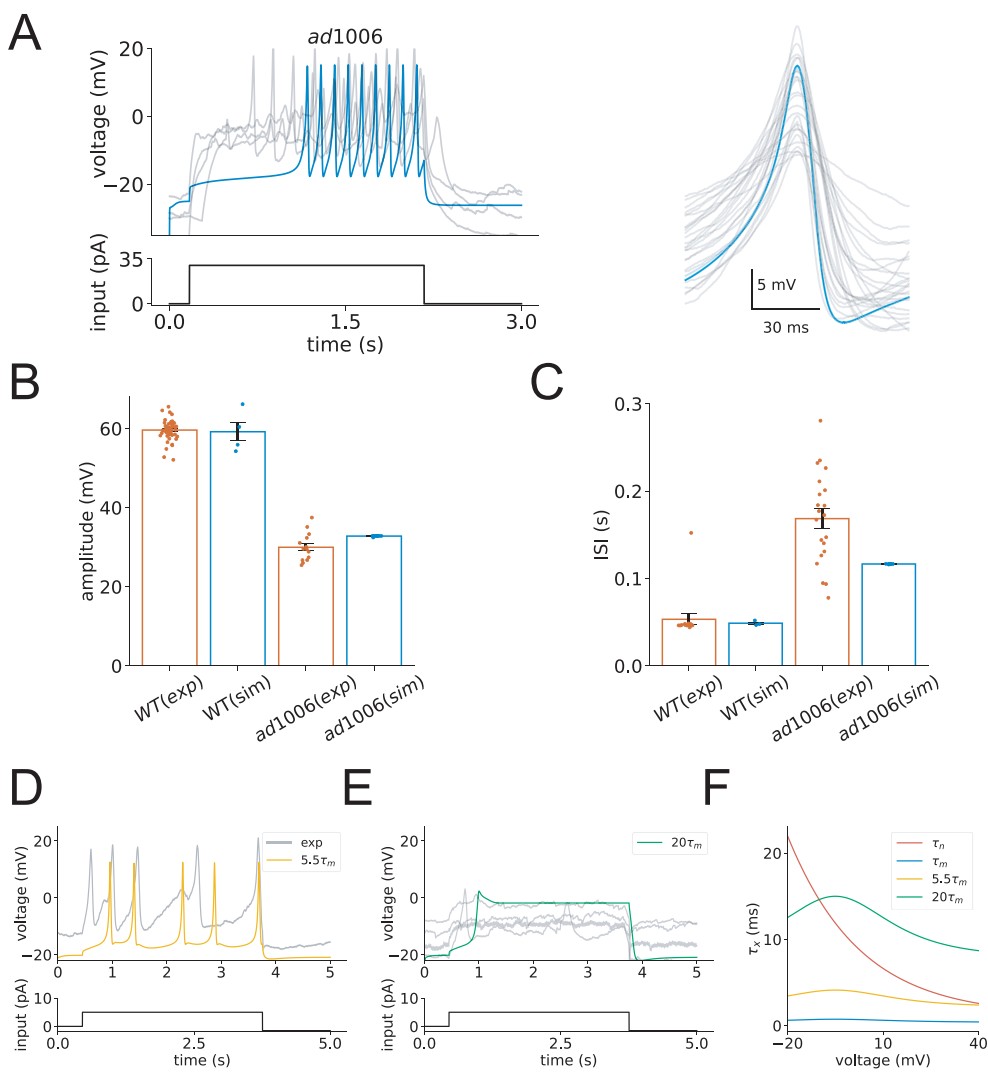

**Fig 5. Analysis of EGL-19 mutant dynamics.** (A) Left panel: Experimental (grey) and simulated (blue) voltage responses of *C. elegans* muscle cells from *egl-19(ad1006,lf)* mutants under a constant input current of 30 pA. Experimental traces from four mutants are shown, demonstrating significant variability among individuals. Right panel: Overlay of individual action potentials from different mutants (grey), aligned with simulated spikes (blue), extracted from the traces on the left. (B-C) Statistical comparison of action potential measurements between wild-type (WT) and *egl-19(ad1006,lf)* mutants. The average amplitude and inter-spike intervals in *egl-19(ad1006,lf)* mutants are significantly different than those in WT. Amplitude (mean $\pm$ SEM): WT(exp), $59.61 \pm 0.35$ mV; WT(sim), $59.19 \pm 2.31$ mV; *ad1006*(exp), $29.98 \pm 0.92$ mV; *ad1006*(sim), $32.81 \pm 0.04$ mV. Inter-spike interval: WT(exp), $53.28 \pm 6.39$ ms; WT(sim), $48.81 \pm 1.29$ ms; *ad1006*(exp), $168.51 \pm 11.22$ ms; *ad1006*(sim), $116.58 \pm 0.12$ ms. The number of animals recorded per genotype: $WT : n = 6$; *ad1006*(exp) : $n = 4$. (D) Comparison of experimental voltage responses from *egl-19(n582,lf)* mutants and simulated data with an altered time constant $\tau_m$ (5.5-fold). (E) Simulated voltage responses with an altered time constant $\tau_m$ (20-fold) compared to *egl-19(n582,lf)* mutants that did not exhibit action potentials. Data from four mutants are displayed in grey, with the simulated trace shown in green. (F) Time constants $\tau_m$ (in blue), $\tau_n$ (in red), with altered $\tau_m$ simulations (5.5-fold in yellow, 20-fold in green).

injection conditions in experiments, we observe that *egl-19(n582,lf)* mutants rarely fire, with only a few neurons sparsely spiking, as shown in Fig 5D and 5E. These results demonstrate that our model accurately reflects the experimental findings.

When substituting extracellular sodium ions ($Na^+$) with N-methyl-D-glucamine (NMDG), notable changes occur in the inter-spike-intervals of muscle cells. In our quest to pinpoint the channels responsible for these observations, we scrutinize the voltage clamp data pertaining to all ion channels involved in action potential generation. The removal of extracellular sodium ions first leads to a cessation of the NCA sodium leak current. Our investigation further reveals that the SLO-2 channel exhibits an increased steady-state value in the absence of $Na^+$ compared to the normal condition. The *slo-2* encodes a subunit of the $K^+$ channel that is modulated by calcium and chloride ion concentrations [39]. The absence of extracellular $Na^+$ induces a shift in the dynamics of the SLO-2 channel, contributing to the altered action potentials. After calibrating the SLO-2 and NCA channel parameters in our model by modifying the maximum conductance of SLO-2 and setting the NCA current to zero, we find that the simulation results closely align with the experimental data, as shown in Fig 6A. A statistical comparison between the simulation and experimental results is provided in Fig 6B and 6C.

### Frequency preferences of body-wall muscle model

The behavioral states in animals are often characterized by network oscillations with specific frequencies, as documented in various studies [59–62]. Neuroscientists have extensively explored how different neuronal types within these networks react to oscillatory inputs, meticulously recording responses to sinusoidal stimuli at preferred frequencies [59]. Building on this foundation, we investigate the response of body-wall muscle cells to oscillatory input patterns.

To provide a comprehensive understanding of these responses, we use a ZAP current as our oscillatory input. The ZAP current is particularly useful because it covers a broad range of frequencies, allowing us to systematically examine how the muscle cells respond to different oscillatory inputs. By applying a ZAP current with a linear frequency sweep from 0.01 to 30 Hz to our model [63], we can observe the cells' behavior across this spectrum.

The ZAP current is governed by

$$I_{ZAP} = I_{max} \sin(2\pi f(t) \cdot t), \tag{12}$$

where $f(t)$ represents the frequency range swept by the ZAP function. For $f(t)$, we utilize a linear chirp function:

$$f(t) = f_{min} + (f_{max} - f_{min}) \cdot t/T. \tag{13}$$

To effectively estimate impedance magnitude, we transform current ($I$) and voltage ($V$) recordings from the time domain into the frequency domain using fast Fourier transforms (FFTs). The impedance ($Z$) is calculated by taking the ratio of the FFT of the voltage to the FFT of the current, as represented by

$$Z = \frac{FFT(V)}{FFT(I)} = Z_{real} + iZ_{imag}. \tag{14}$$

The impedance magnitude is then expressed as a function of frequency, forming an Impedance–Magnitude (IM) profile, as illustrated in Fig 7B. Notably, the body-wall muscle cells of *C. elegans* exhibit a distinct preferred frequency at around 4.7 Hz. To compare with experimental data, we perform a statistical analysis of inter-spike intervals across 10 spike trains recorded from three wild-type *C. elegans* individuals, as shown in S5 Fig. While the distribution is not perfectly bimodal, there is a noticeable gap in the distribution. We set a 200

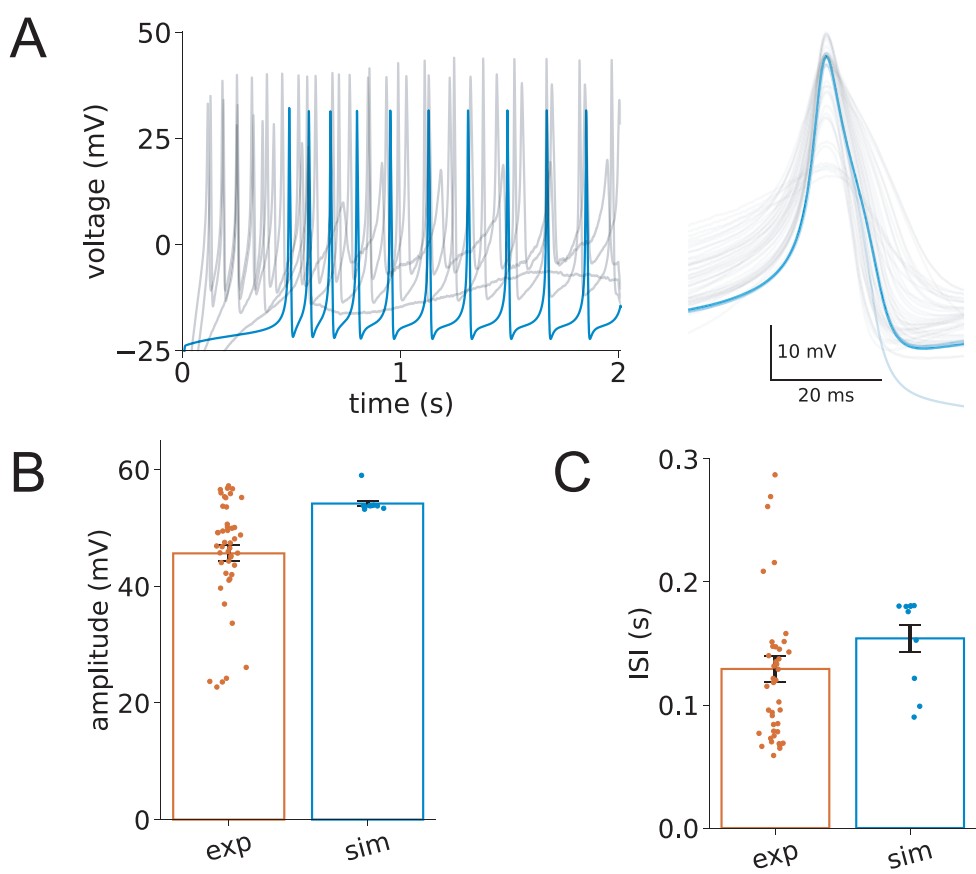

**Fig 6**. **Investigating the impact of potassium channel SLO-2 on neuronal dynamics in a sodium-ion-free environment.** (A) Left panel: Voltage traces recorded experimentally (gray) from WT in response to a 30 pA input current applied for 2 seconds, alongside corresponding simulated traces (blue) in a sodium-ion-free solution. Right panel: Overlay of individual action potentials (grey), aligned with simulated spikes (blue), extracted from the traces on left. (B,C) Statistical comparison between experimental (exp) and simulated (sim) data. Amplitude: exp, 45.66 ± 1.37 mV; sim, 54.55 ± 0.79 mV. Inter-spike-interval: exp, 129.33 ± 10.30 ms; sim, 154.17 ± 11.01 ms. The number of animals recorded: $n$ = 4. Error bars represent the standard error of the mean (SEM).

ms threshold based on this separation, providing a reasonable cutoff to differentiate between burst and regular firing patterns. Additionally, we apply an extra criterion to define burst activity based on the reference [64]: a burst event is accompanied by a sustained depolarization phase (platform phase) with a voltage above –18 mV, as shown in Fig 7C. The choice of –18 mV is based on our analysis of the afterhyperpolarization (AHP) trough potential distribution of each spike, as shown in S5 Fig. Therefore, in a burst event, the burst duration is defined as the period between the membrane potential rising above –18 mV before the first spike and decreasing back to –18 mV after the last spike. In cases where the membrane potential consistently remains above –18 mV, the burst duration is determined by the inflection point prior to the first spike as the onset and the inflection point following the last spike as the offset. This dual criterion—combining ISI and depolarization plateau—ensures a precise identification of burst events, distinguishing them from random spike occurrences. Ultimately, we find that the preferred frequency of around 4.7 Hz, corresponds to a burst firing mode, as depicted in Fig 7C. These high-frequency bursts are critical for generating the necessary force

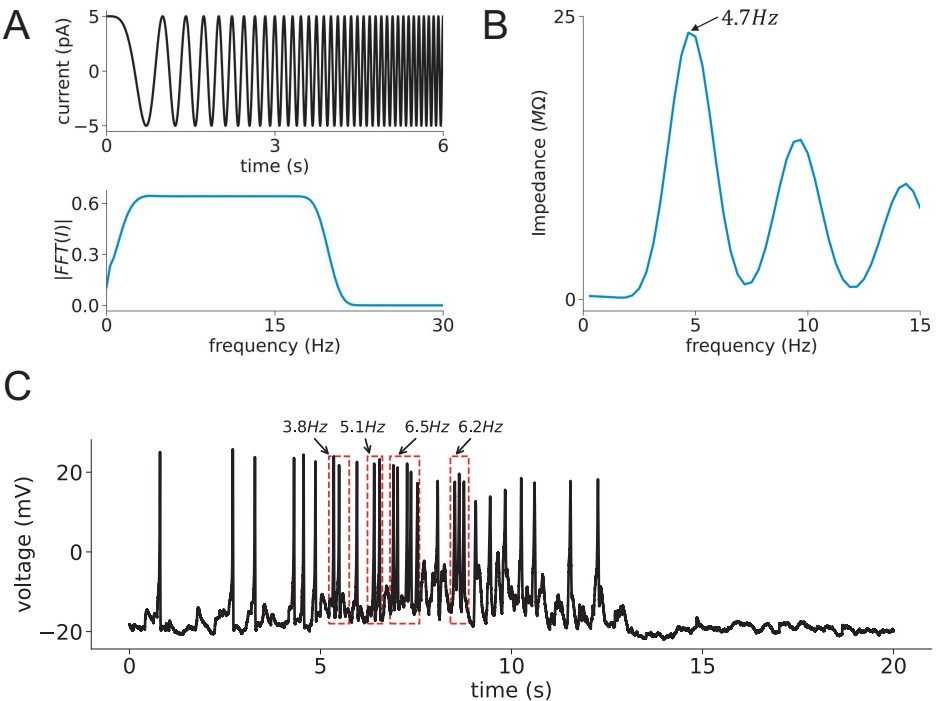

**Fig 7**. **Analyzing frequency preferences in the body-wall muscle cell model.** (A) The 100-second ZAP current injection (represented by the black trace) is applied to the body-wall muscle cell model with a constant amplitude and a linearly varying frequency. The corresponding power spectra are shown in the lower panel, indicated by the blue line. (B) The graph of impedance magnitude as a function of frequency for the body-wall muscle cells, determined by the ratio $FFT(V)/FFT(I)$. (C) A representative spontaneous action potential curve in WT cells, showing both "burst" and "regular" firing modes. The firing rates of cells in the "burst" firing mode are calculated. Statistical analysis of data obtained from 3 WT cells shows a "burst" mode frequency of $4.8 \pm 1.05$ Hz and a "regular" firing mode frequency below 2 Hz.

and coordination for effective movement. While the undulation frequency of *C. elegans* during locomotion is barely exceeds 2 Hz for swimming animals and is even slower during crawling on agar plates, these observed undulation frequencies do not directly correspond to the firing rates of individual muscle action potentials. Instead, these behaviors are driven by the coordinated activity of multiple muscle groups, leading to rhythmic, clustered action potential firing within individual muscle cells. These rhythmic bursts can occur at higher frequencies, ranging from 3.4 Hz to 6.5 Hz [65,66]. Moreover, the burst firing is often driven by plateau potentials in motor neurons, which modulate neurotransmitter release in a graded manner, enabling sustained and synchronized muscle activation during locomotion [40,67].

## Discussion

This study advances the understanding of the neurophysiology of *C. elegans* body-wall muscle cells by integrating detailed computational models with empirical data analysis. Our biophysical model effectively captures the main features of electrical dynamics in wild-type cells. Moreover, it predicts alterations in the dynamic properties of *C. elegans* body-wall muscle cells across various mutants and in sodium-ion-free solutions. Our work also provides a parallel SBI algorithm using GPU vectorization and parallelization, which allows for extensive and efficient exploration of the model's parameter space. Notably, our algorithm can be

scaled to simulate not just individual neurons or muscles, but networks of cells, as detailed in the S2 Appendix. Additionally, by linking model dynamics with physiological functions, we identify a distinct preferred frequency in *C. elegans* body-wall muscle cells. This optimal frequency induces a burst firing mode, which may significantly enhance the force of muscle contractions.

To improve the performance of our algorithm, we integrate both experimental and numerical approaches. Experimentally, we conduct investigations by reviewing the literature [23,42, 68,69] and employing gene manipulation and electrophysiology experiments. Accordingly, we identify the ion channels that influence the dynamics of body-wall muscle cells and select the key channels to incorporate into our model. From the algorithmic standpoint, we implement three strategies to enhance convergence and improve the accuracy of parameter estimation. First, we initialize prior distributions that encompass the full physiological range of ion channel kinetics, enabling the model to explore a sufficiently broad parameter space. Second, we estimate certain parameters for each ion channel separately based on voltage-clamp experimental data. Consequently, we only need to estimate the rest few parameters for each ion channel, thereby facilitating algorithm convergence. Lastly, we develop effective statistics to capture the dynamics we aim to replicate. These statistics—including action potential counts, latency to the first action potential, mean and variance of membrane potentials, mean resting membrane potentials, and mean inter-spike intervals—serve to reduce the complexity of the high-dimensional output while preserving key electrophysiological characteristics.

In studying the locomotion of *C. elegans*, experimental findings indicate that their movement is regulated by a network of excitatory cholinergic (A- and B-types) and inhibitory GABAergic (D-type) motor neurons along the nerve cord, which innervate the muscle cells lining the worm's body [29,70]. Additionally, a series of interneurons indirectly regulate the worm's movement by modulating motor neurons and proprioception [29,71]. Current research predominantly focuses on modeling the neurons involved and investigating the underlying physiological mechanisms, with comparatively less emphasis on body-wall muscle cells [72–76]. However, muscle cells play a crucial role in locomotion as they integrate neuronal inputs and deliver all-or-nothing electrical outputs to drive movement. The biophysically detailed model we developed characterizes the physiological mechanisms underlying body-wall muscle cells, providing insights for further exploration of the interactions between motor neurons and muscle cells. On the other hand, our modeling framework can serve as a solid foundation for future explorations in modeling other neurons within *C. elegans*. Previous work on neuron model parameter estimation, such as those conducted on various types of neurons in the mouse visual cortex by the Allen Brain Project [77] and on neuron models in the electric fish *Apteronotus* [78], has employed parameter estimation methods based on several evolutionary algorithms, including Differential Evolution, Dual Annealing, and Particle Swarm Optimization [79–81]. We apply all these algorithms to the parameter tuning of our model and compare their running times, as shown in S4 Fig. Specifically, in the task illustrated in Fig 4E, we estimate 6-dimensional parameters and set the algorithms to stop when the relative error fell below 0.01. These algorithms converge more slowly, often taking several hours and requiring hundreds of iterations to stop, which is consistent with previous studies [82,83]. In contrast, our method demonstrates significantly faster convergence, requiring only 3 iterations, representing more than an order-of-magnitude improvement in speed. Additionally, our method provides a probabilistic framework for estimating uncertainty in parameter inference. This is beneficial in the presence of biological variability and experimental noise, as demonstrated by the variability observed among body-wall muscle cells in this study.

In conclusion, our biophysical model presented here may shed light on the underlying mechanisms for electrical activities in *C. elegans* body-wall muscle cells and offer a generalized framework for detailed modeling in the study of the *C. elegans* locomotion system. Future directions may include utilizing this modeling framework to develop detailed biophysical models for various motor neurons within the *C. elegans* motor circuits. This will enable a more thorough investigation into the interactions between motor neurons and muscle cells during locomotion, enhancing our understanding of the system's complexity.

## Supporting information

**S1 Fig. Voltage clamp currents of the additional potassium channels.** The top section of each column illustrates the currents in different mutants, while the bottom section depicts the ionic currents obtained by subtracting the corresponding mutTIF currents from the total wild type (WT) currents. Only nominal alterations are observed in these two mutants, *shl-1(lf)* and *slo-1(lf)*.
(TIF)

**S2 Fig. Elicited spike trains across varying stimulation currents with four primary ion channels.** (A-D) These figures present elicited spike trains under varying stimulation currents, ranging from 15 pA to 30 pA in increments of 5 pA. The simulation results are shown in blue, compared to red curves representing experimental data.
(TIF)

**S3 Fig. Posterior distribution.** The corner plot displays the marginal and pairwise marginal distributions of the 6-dimensional posterior over gap junction parameters $J_{12}, J_{23}, \ldots, J_{56}$. The true parameter values, marked by red lines, are successfully captured within the high-probability regions of the posterior distribution.
(TIF)

**S4 Fig. Performance comparison of four optimization methods.** (A) Comparison of running time (in minutes) among four optimization methods for solving the same task in Fig 4E: parallel simulation-based inference (P-SBI), dual annealing (DA), differential evolution (DE), and particle swarm optimization (PSO). The bars represent the average running time across 10 trials, with error bars indicating the standard deviations. (B) Relative error convergence of four optimization methods, plotted as a function of the logarithm of iterations. The relative error is calculated as $\frac{||\mathbf{x}_o - \hat{\mathbf{x}}_o||_2}{||\mathbf{x}_o||_2}$, where $\mathbf{x}_o$ is the target solution and $\hat{\mathbf{x}}_o$ is the estimation. The algorithms stop when the relative error falls below 0.01; P-SBI achieves a solution with a relative error less than 0.01 in just 3 iterations.
(TIF)

**S5 Fig. Spike train analysis of body-wall muscle cells.** (A) This panel shows the inter-spike interval (ISI) distribution derived from spike trains of multiple body-wall muscle cells. The x-axis represents the ISI in seconds, and the y-axis indicates the number of occurrences for each interval. (B) This panel presents the distribution of afterhyperpolarization (AHP) trough potentials derived from the same spike trains. The x-axis represents the membrane potential (mV), while the y-axis shows the frequency of occurrences within each potential range.
(TIF)

**S1 Appendix. Equations used in the model simulations.**
(PDF)

**S2 Appendix. Network model parameter tuning.**
(PDF)

**S1 Table. Model parameters.**
(PDF)

**S1 Data. The relevant experimental data in the paper.**
(ZIP)

## Author contributions

**Conceptualization:** Xuexing Du, Yunzhu Shi, Shangbang Gao, Douglas Zhou.

**Data curation:** Yunzhu Shi, Shangbang Gao.

**Formal analysis:** Xuexing Du, Jennifer Crodelle, Victor James Barranca, Songting Li, Douglas Zhou.

**Funding acquisition:** Songting Li, Shangbang Gao, Douglas Zhou.

**Investigation:** Xuexing Du, Songting Li, Douglas Zhou.

**Methodology:** Xuexing Du, Songting Li, Shangbang Gao, Douglas Zhou.

**Project administration:** Songting Li, Shangbang Gao, Douglas Zhou.

**Resources:** Yunzhu Shi, Shangbang Gao.

**Software:** Xuexing Du, Songting Li, Douglas Zhou.

**Supervision:** Jennifer Crodelle, Victor James Barranca, Songting Li, Shangbang Gao, Douglas Zhou.

**Validation:** Xuexing Du, Jennifer Crodelle, Victor James Barranca, Songting Li, Shangbang Gao, Douglas Zhou.

**Visualization:** Xuexing Du, Songting Li, Douglas Zhou.

**Writing – original draft:** Xuexing Du, Songting Li, Yunzhu Shi, Shangbang Gao, Douglas Zhou.

**Writing – review & editing:** Xuexing Du, Jennifer Crodelle, Victor James Barranca, Songting Li, Yunzhu Shi, Shangbang Gao, Douglas Zhou.

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
