## [Decision Letter · Decision Letter 0]

12 Sep 2024

Dear Mr. Du,

Thank you very much for submitting your manuscript "Biophysical Modeling and Experimental Analysis of the Dynamics of C. elegans Body-Wall Muscle Cells" for consideration at PLOS Computational Biology.

As with all papers reviewed by the journal, your manuscript was reviewed by members of the editorial board and by several independent reviewers. In light of the reviews (below this email), we would like to invite the resubmission of a significantly-revised version that takes into account the reviewers' comments.

Both reviewers value the topic and scope of this paper. However, the key methodology implemented could be better explained and the limitations and advantages of the computational algorithm discussed. Moreover, the results could be related back to the experimental observations, by making a comparison between the measured and simulated action potentials (APs) between the mutants and to the phenotypes and the effects on muscle function.

We cannot make any decision about publication until we have seen the revised manuscript and your response to the reviewers' comments. Your revised manuscript is also likely to be sent to reviewers for further evaluation.

Sincerely,

Fleur Zeldenrust

Academic Editor

PLOS Computational Biology

Marieke van Vugt

Section Editor

PLOS Computational Biology

Both reviewers value the topic and scope of this paper. However, the key methodology implemented could be better explained and the limitations and advantages of the computational algorithm discussed. Moreover, the results could be related back to the experimental observations, by making a comparison between the measured and simulated action potentials (APs) between the mutants and to the phenotypes and the effects on muscle function.

Reviewer's Responses to Questions

**Comments to the Authors:**

Reviewer #1: This is a solid and well written computational study to model the electrophysiological property of muscle cells of C. elegans, which exhibit a calcium-mediated action potential. The Hodgkin-Huxley-type simulation of voltage gated current and membrane potential dynamics is straightforward and has been used in many prior works. The main development of the current study is to leverage the modern deep learning method and high-speed parallel computing for efficient parameter search. The simulated results appear to agree well with experimental measurements. However, the key methodology implemented in this work is not clearly explained and investigated, making it difficult for readers to understand the limitations and advantages of their computational algorithm.

Major concern:

1. I do not understand the parallel simulation-based inference algorithm, because the rationale underlying MAP is not explained. The inference neural network was trained on simulated data. However, there is no guarantee that the simulated data would in any way resemble the experimentally measured neural dynamics: the fundamental biophysical ingredients of your model and the prior distribution of the parameters play a critical role here. It is possible that no matter how many simulations you run, the resulting dynamics {x_i} remain qualitatively different from the observed data. Related to this point, the convergence of the posterior distribution conditioned on data (p(\theta|x_0) is a necessary but not sufficient condition for finding the correct solution space. In fact, under what scenario would this algorithm actually converge?

Other concerns:

2. This work will make a larger impact if the author could discuss whether the current method can be scaled up to simulate not just a single neuron/muscle, but a network of cells. It is also not clear to me when comparing with other methods such as evolutionary algorithms, what is the main advantage of the current method? If speed is a major bottleneck, could you provide a quantitative comparison based on existing literature reports?

3. Related to my major concern. Figure 4E appears to suggest that the posterior is modelled as the multivariate Gaussian distribution. Is this an assumption underlying your model?

4. The simulation (Figure 7) predicts burst firing for periodic inputs with a frequency > 4Hz. C. elegans undulation frequency barely exceeds 2Hz for swimming animals, and the worm crawls much slower on agar plate. Thus, this prediction does not appear to be behaviorally relevant. Do the author suggest that motor neurons would fire at a high frequency in order to strongly active the muscle cells? They should make their points clear here.

Reviewer #2: This study presents a biophysical model of C. elegans muscle action potentials as well as introducing a new method of parameter fitting for the model. Both are potentially useful for the field. As my expertise lies in C. elegans neuroscience I have focused my comments there. In general quantitative comparison between measured and simulated action potentials (APs) would help support specific conclusions drawn in the text. In addition, relating the findings particularly about the mutants back to the phenotypes and effects on muscle function in a concrete way would illustrate the power of the student and improves its impact.

Specific Comments:

Much of the first paragraph does not seem relevant. Creation of human-level intelligent system seems to be a far cry from modeling muscle excitation. Much of the value of the C. elegans stems from analysis of specific genes/constructs mutations in a simple system. A comprehensive biophysical model of muscle activation will be very helpful in this regard. Thus, the focus, in my opinion is more centered on modeling of individual cell activity to study cell elements involved as well a modeling/understanding the simple locomotory systems in C. elegans.

The trace in Fig 1a) is referred to as a graphical representation? This is actual an experimental measurement, correct? That should be explained explicitly. “Representation” is vague. This is simply a baseline measurement of membrane potential in a wild-type animal? Again please explain briefly in the main text. Do the muscles twitch when they spike? There are no motion artifacts? Or how are they compensated for?

Fig1 The two egl-19 lf mutants show different effects on calcium current. Do they have differing phenotypes in terms of effects on muscle function or are they similar? See point in discussion below.

Pg 9: “We note that when the model contains only the four previously mentioned ion channels, the action potentials exhibit premature repolarization compared to experimental data.” Is that shown some place? Would be good to illustrate the difference for the reader can fully understand.

pg 14 “Specifically, under a 30 pA current injection, the average action potential amplitude in wild-type cells is 61.6 mV, as shown by the red curve in Fig. 4A. In contrast, in the egl-19(ad1006,lf) mutants, the amplitude is less than half of this value, as depicted by the red curve in Fig. 5A.”

This is an observation of individual experimental trials. Measurement of numbers action potential and comparison of mean would be more convincing.

Pg 14 “Notably, the mutant model agrees with the shape of action potentials under 340

constant current injections, as shown in Fig. 5B.”

I do not see where the model is compared to the actual data. Overlaying the model trace on top of a number of measured action potentials (or the means of a number of action potentials) would be more convincing. Likewise of measurement of AP frequency comparing measured vs simulated would be more convincing.

Comparing Fig 5C and D simulations of action potentials in elg-19(n582,lf) do not match the shape of measured action potentials in experiments. Again overlaying the simulated AP with a number of experimental measured APs would add in this comparison. This is not commented on at all and should be addressed. Why doesn’t it match the AP profile? what else might be going on here? The simulation does appear to predict the slower firing rate but again there is no quantitative comparison of measured firing rate vs simulated firing rate.

Fig 6 Again comparison of AP via with overlayed individual AP traces and quantitative measurement of AP frequency would be helpful.

Pg 17 How is the ISI threshold of burst set to 200ms? Is there a clear bimodal distribution in ISI values that clearly delineate burst from normal firing or is it a continual distribution with an arbitrary threshold? This needs to be explained more clearly and justified.

Pg 17 The measured burst frequency of 4.8 Hz is surprisingly close to the burst cutoff time of 200 ms (i.e. 5Hz, although I also don’t fully understand how it could be lower than 5Hz?). Does the burst frequency depend on the ISI burst threshold time? It seems like it might or needs to be shown otherwise. If so a very strong justification of the threshold time is needed. Without further explanation of the burst frequency measurement, claims about the optimal response frequency of of muscles cells cannot be justified.

Discussion: An advantage of C. elegans is it genetic flexibility to disrupt individual genes etc.. in a simple controlled system. As mentioned the modeling of muscle function presented here could be valuable for assessing effects of specific mutants etc.. but there is little attempt to tie the results back to actual muscle function/behavior in a meaningful way. For example, the two egl-19lf mutants show different effects on Ca current that can be understood and explored through the model. How do crawling/muscle function phenotypes compare between these two alleles and how does that relate to what has been demonstrated on a physiological level? A discussion such as this would add weight to the findings and study.

**Have the authors made all data and (if applicable) computational code underlying the findings in their manuscript fully available?**

Reviewer #1: Yes

Reviewer #2: Yes

PLOS authors have the option to publish the peer review history of their article (what does this mean?). If published, this will include your full peer review and any attached files.

Reviewer #1: No

Reviewer #2: **Yes: **Christopher Gabel
---

## [Decision Letter · Decision Letter 1]

7 Jan 2025

Dear Mr. Du,

We are pleased to inform you that your manuscript 'Biophysical Modeling and Experimental Analysis of the Dynamics of C. elegans Body-Wall Muscle Cells' has been provisionally accepted for publication in PLOS Computational Biology.

Best regards,

Fleur Zeldenrust

Academic Editor

PLOS Computational Biology

Marieke van Vugt

Section Editor

PLOS Computational Biology

Reviewer's Responses to Questions

**Comments to the Authors:**

Reviewer #1: The authors have satisfactorily addressed my questions.

Reviewer #2: Authors have done good job responding to the earlier reviews. One final comment: The addition of the quantification in Fig 5 B,C and Fig 6 B,C is very helpful. Are there statistical test used to demonstrate the significance in the differences between WT and Mutant data sets, and conversely no significant differences between the experimental and stimulated data sets? The results are obvious by eye but statistical test would add more evidence.

**Have the authors made all data and (if applicable) computational code underlying the findings in their manuscript fully available?**

Reviewer #1: None

Reviewer #2: Yes

PLOS authors have the option to publish the peer review history of their article (what does this mean?). If published, this will include your full peer review and any attached files.

Reviewer #1: No

Reviewer #2: **Yes: **Christopher Gabel

---

## [Editor Report · Acceptance letter]

PCOMPBIOL-D-24-01178R1

Biophysical Modeling and Experimental Analysis of the Dynamics of C. elegans Body-Wall Muscle Cells

Dear Dr Du,

I am pleased to inform you that your manuscript has been formally accepted for publication in PLOS Computational Biology. Your manuscript is now with our production department and you will be notified of the publication date in due course.

With kind regards,

Zsofia Freund
